# Numerical Simulation of the Pouring Process in Deep-Water Flexible Pipe End Fittings

**DOI:** 10.3390/polym15061397

**Published:** 2023-03-10

**Authors:** Hongxia Li, Chenwei He, Tao Zhang, Qingzhen Lu, Jun Yan

**Affiliations:** 1Department of Mechanical Engineering, Dalian University of Technology, Dalian 116024, China; 2State Key Laboratory of Structural Analysis for Industrial Equipment, Department of Engineering Mechanics, Dalian University of Technology, Dalian 116024, China

**Keywords:** flexible pipe, end fitting, epoxy resin, numerical simulations

## Abstract

The pouring quality of epoxy resin in the resin cavity of deep-water composite flexible pipe end fittings has a decisive effect on the performance of the end fitting, and accurate investigation of the resin flow during the pouring process can provide a reference for optimization of the pouring process so that the pouring quality can be improved. In this paper, numerical methods were employed to investigate the pouring process of the resin cavity. The distribution and evolution of defects were studied, and the influence of pouring rate and fluid viscosity on pouring quality was analyzed. In addition, based on the simulation results obtained, some local pouring simulations were conducted for the armor steel wire, the key characteristic structure of the end fitting resin cavity that has a significant influence on the pouring quality, to investigate the influence of the geometrical characteristics of the armor steel wire on the pouring quality. Based on these results, the existing end fitting resin cavity structure and pouring process were optimized, and better pouring quality was obtained.

## 1. Introduction

In recent years, with the continuous development of fossil resources, increasing attention has been given to the development of offshore fossil resources [1]. Flexible pipes are indispensable transportation equipment in the exploitation of offshore oil and gas re-sources [2]. Deep water environments have high requirements for the performance of flexible pipes; however, the manufacturing of high-performance composite flexible pipes still faces some difficulties.

The type of high-performance composite flexible pipe most commonly used in deep water is a metal-armored unbonded flexible pipe [3]. This type of flexible pipe consists of four key components: metal interlocking carcass layers, metal armor layers, inner and outer sheath layers and an end fitting. As a key component, the end fitting must not only achieve a reliable connection with the armor layers but also good internal and external sealing [4]. To achieve good connection, a special epoxy resin casting method is generally adopted when forming the end connection [5], and the wedge ring contact method is used to achieve sealing [6]. The typical structure of a flexible pipe end fitting is shown in Figure 1 [7]. One of the main problems that must be solved during the manufacturing of the end fitting is the problem of low filling rate. Due to the special structure of the end fitting, the vacuum casting process that is traditionally used with epoxy resin cannot create a full vacuum within the end fitting resin cavity during the pouring process, and it therefore cannot ensure the low porosity of the epoxy resin material after pouring. On the other hand, the presence of a large number of tensile armor steel wires in the end fitting resin cavity makes the pouring path complicated; as a result, cavities, bubbles and other defects are readily formed, and this contributes to the low filling rate. The end fitting is connected by the resin casting method, and the bonding strength of the epoxy resin to the armor steel wires in the resin cavity directly affects the performance of the end fitting. Incomplete filling of the resin cavity will lead to a decrease in the bonding strength of the epoxy resin and armor steel wires, and the performance of the end fitting will not reach the standard. Therefore, it is necessary to improve the existing resin cavity pouring process of the end fitting.

At present, there are relatively few studies of the epoxy resin part inside the end fit-ting; only two main aspects have been studied. One is the tensile capacity of the tensile armor steel wires bonded with epoxy resin inside the end fitting. Aiming at prolonging fatigue life, Xavier et al. designed a new anchorage model for tensile armor steel wires in the end fitting resin cavity of a flexible pipe and, using the resin and stress concentration factor, compared its stress with those of two other commercial models by finite element simulations and experimental tests on a reduced scale; the results demonstrated the superiority of the model. The prototype test of the new model was then performed on an actual scale, proving the feasibility of the new model in prolonging the fatigue life [8]. Based on Xavier’s work, Bueno established a three-dimensional finite element model of the connection between an unbonded flexible pipe and an end fitting and studied the fatigue behavior of the tensile armor steel wires in comparative experiments [9]. Filho et al. established a finite element model of resin armor steel wire anchorage in an end fitting considering friction and thermal effects and found via experiments that the model was better than a model that took into account only the cohesive condition [10]. Campello et al. compared a novel flexible pipe end fitting designed by Petrobras with the current end fitting used by the industry with respect to stress distribution along the steel wire inside the end fitting and fatigue performance. Finite element simulations were conducted in which the entire end fitting mounting process and the operational loads were considered, and the results show that use of the novel structure results in a considerable improvement in end fitting fatigue performance [11]. Anastasiadis et al. examined the stress concentration and the stress concentration factors (SCF) of the armor steel wires inside the end fitting under different axial tensions using two-dimensional finite element simulation. A parametric study was also performed to identify potential factors that might affect the SCF values. Based on the results, an approximate relationship that can be used to estimate the maximum SCF on the steel wire inside the end fitting was established [12]. Yijun Shen et al. divided the failure process of a flexible pipe end fitting into three sections: complete bonding, semipeeling, and complete peeling, and they analyzed the stress distribution of the tensile armor layer in each process [13]. Based on the above literature, Ireland’s Adrian Connaire et al. used the cohesive element of ABAQUS to simulate the adhesive layer between the tensile armor layer and the epoxy resin and studied the effect of adjustment of the ultimate stress on the finite element results [14].

Another aspect that has been studied is the resin vacuum casting process. Zhang et al. designed an asymmetric eccentric large blade agitator and changed the stirring speed to enhance the stirring effect, effectively solving the problem that polyurethane resin materials are difficult to mix evenly in the vacuum pouring process [15]. Bin Ahmad et al. optimized the three parameters of resin temperature, mold temperature and vacuum pressure time in the vacuum casting process by the Taguchi method and were able to reduce the shrinkage effect by 52.4% [16]. Jin Tao et al. simulated the curing process of epoxy resin casting transformer coils and applied the temperature control method to reduce the problem of crack formation due to stress concentration in the epoxy resin casting electrical components [17]. Fan Xiaohong et al. analyzed the factors that affect product performance and appearance quality in the vacuum casting process of epoxy resin and proposed methods to solve appearance quality problems [18]. Wu Haijun studied the process of and precautions related to the epoxy resin vacuum casting process [19].

In the literature, little attention has been given to the pouring process of the end fitting resin cavity. However, an accurate understanding of how resin flow occurs during the pouring process can provide a reference for optimization of the epoxy resin pouring process for end fitting. Therefore, it is necessary to study the resin flow in the pouring process of the end fitting epoxy resin cavity. In terms of geometric shape, the resin cavity of the end fitting is a short, thick cavity with a complex structure; it is therefore difficult to clearly observe the resin flow inside the cavity, even if a transparent mold is used in pouring experiments. Considering the experimental cost, numerical simulation methods are more suitable for this type of study. In this paper, the resin pouring process of the end fitting resin cavity was studied using a numerical simulation method, and the distribution and the process of evolution of cavities, bubbles and other defects in the end fitting resin cavity were analyzed. In addition, the effects of process conditions such as pouring rate, fluid viscosity, and the local structure of the cavity on pouring quality were studied.

## 2. Methods

### 2.1. Geometry and Finite Element Models

The geometry of the end fitting studied in this paper is shown in Figure 2a, and the model of the resin cavity obtained by cavity extraction is shown in Figure 2b. The axial size of the resin chamber model is 260 mm, and the maximum diameter is 220 mm. The cross section of the two layers of armor steel wire is a rectangle with dimensions of 1.5 mm × 5 mm, and the helix angle is ±35°. The inner armor layer contains 26 steel wires, and the outer armor layer contains 30 steel wires. The bending pattern of the steel wire is shown in Figure 2c; the length marked in the figure is the length along the axial direction of the end fitting.

The resin cavity model was divided into tetrahedral meshes by using finite element preprocessing software Altair Hypermesh. The finite element model is shown in Figure 3; the location of the injection port is indicated in the figure.

### 2.2. Boundary Conditions

The pouring process was simulated by using the finite element simulation software Autodesk Moldflow. According to the actual process, the gate diameter was 25 mm. The mold material was tool steel P-20, and the wall surface was a nonslip surface. The effect of gravity was taken into account; the direction of gravitational force was along the y axis.

In the pouring process of the resin cavity, the resin is in contact with the sheath mate-rial in the end fitting. Because the mechanical properties of the sheath material change at high temperature, it is usually necessary to perform the resin pouring process at room temperature [20]. Accordingly, the mold temperature and the fluid temperature in the simulation were both set to 25 °C. Similarly, according to the actual process, the rate of pouring of the resin was set at 120 cm^3^/s.

In this paper, the reactive viscosity model is used to describe the rheological behavior of the resin. The reactive viscosity model is expressed as follows:(1)$$\left\{\begin{array}{ll} \eta =\eta_{m}{\left(\frac{\alpha_{g}}{\alpha_{g}-\alpha }\right)}^{\left(C_{1}+C_{2}\alpha \right)} & ,\alpha <\alpha_{g}\\ \eta =+\infty , & \alpha \geq \alpha_{g} \end{array}\right.,\;\eta_{m}=\frac{\eta_{0}}{1+{\left(\frac{\eta_{0}\gamma }{\tau^{*}}\right)}^{\left(1-n\right)}},\;\eta_{0}=Be^{\frac{T_{b}}{T}}$$
where *η* is the viscosity (Pa∙s), *γ* is the shear rate (s^−1^), *T* is the temperature (K), *α* is the degree of cure (0~1), and *α_g_*, *C*_1_, *C*_2_, *τ*^*^, *n*, *B*, and *T_b_* are the model parameters. The three equations shown in (1) describe the effects of the degree of cure, the shear rate and the temperature on the viscosity. According to the viscosity properties of Alike E44, the epoxy resin material used in the end fitting resin cavity filling experiment, the values of the individual parameters were 0.5273, 4.09, 1.55 × 10^−5^, 1.31 × 10^−7^ Pa, 0.8294, 7 × 10^−9^ Pa∙s, and 7983 K, respectively, and the reference resin viscosity was 50 Pa∙s at a reference shear rate of 1 s^−1^.

Based on the above boundary conditions, the pouring rate and the resin viscosity were varied, and the effects of these changes were studied. The pouring rates were 40 cm^3^/s, 80 cm^3^/s, 120 cm^3^/s, and 160 cm^3^/s, and the resin viscosities were 30 Pa∙s, 40 Pa∙s, 50 Pa∙s, and 60 Pa∙s. The epoxy resin hardly cures during the pouring process at room temperature, so the effect of resin curing during the pouring process is negligible. The small heat release that occurs during resin curing and use of the same temperature for the mold and the resin cause the temperature of the resin to not change significantly during the pouring process; therefore, the influence of changes in temperature during the pouring process can also be ignored. However, the change in viscosity caused by the change in shear rate is not negligible. Therefore, when the resin viscosity is varied to study the influence of viscosity on pouring quality, the influence of the shear rate on the viscosity should remain constant; this means that the parameters *τ** and *n* should remain un-changed. The values used to modify *B* were 3.786 × 10^−9^ Pa∙s, 5.35 × 10^−9^ Pa∙s, 7 × 10^−9^ Pa∙s, and 8.708 × 10^−9^ Pa∙s.

### 2.3. Mesh Independency

Based on the above models and boundary conditions, a mesh independency test was performed by using four element models with four maximum element sizes of 10 mm, 7.5 mm, 5 mm, and 3.5 mm. Variation in the number of air trap locations with the maximum element size was studied. As shown in Figure 4, the number of air trap locations of the grids with the maximum mesh sizes of 5 mm and 3.5 mm are almost unchanged. In order to balance a reasonable computational cost with acceptable results, a maximum element size of 5 mm was used.

## 3. Results and Discussion

### 3.1. Simulation Result and Analysis

#### 3.1.1. Distribution of Air Traps

Figure 5 shows the air trap results obtained in the pouring simulation when a pouring rate of 120 cm^3^/s and a resin viscosity of 50 Pa∙s were used. As shown in the figure, most of the air traps are distributed on the surface of the armor steel wires; Figure 5b shows that most of the air traps are located at positions at which the two layers of armor steel wires overlap. This is due to the presence of small gaps between the armor steel wires at the overlapping positions that make it difficult for the resin to flow into and fill the space. For the same reason, on the right side of the resin cavity in the area where the end fitting connects to the main body of the flexible pipe, air traps frequently appear in the position where the armor steel wires are gathered, as shown in Figure 5d. In the axial direction, compared with the two ends of the cavity, there are fewer air traps in the middle of the cavity. This is because the armor steel wires in the middle of the cavity bulge outward, and the gaps between the steel wires are relatively large; thus, it is relatively easier for the resin to flow into and fill these spaces. In addition, Figure 5a,c show that there are also a small number of air traps at the places where the armor steel wires bend and at the corners of the cavity wall. It can be seen that eliminating sharp corners in the cavity is beneficial to filling the cavity.

#### 3.1.2. Resin Flow Condition

Figure 6 shows the simulated resin pouring process when a pouring rate of 120 cm^3^/s and a resin viscosity of 50 Pa∙s are used. The total resin pouring process requires 71.7 s. The appearance of the end fitting at nine time points during this period is shown in the figure. As shown in the figure, although there are gaps between the armor steel wires, due to the high viscosity of the resin, part of the resin exhibits a phenomenon similar to that seen when resin flows over the surface of a solid cylinder during pouring. The other part of the resin passes through the gaps between the armor steel wires and flows below the armor steel wires. Between the two parts of the resin, there is an air layer. When the resin flows in the axial direction, the upper resin layer may obstruct the discharge of the air be-low, resulting in the appearance of air traps. Comparison of the appearance of the end fit-ting cavity at 62.920 s and 71.700 s shows that resin filling of the space at the sharp corner on the pipe side of the resin cavity is difficult, indicating that such structures should be avoided in the design of the end fitting resin cavity. This is consistent with the conclusion obtained from the analysis of the air trap results. On the pipe side of the resin cavity, resin filling is also difficult in the position where the armor steel wires are gathered, a finding that is also consistent with the previous results obtained regarding the formation of air traps. Moreover, in terms of the sequence in which the resin fills the resin cavity, setting an exhaust vent above the pipe side of the cavity facilitates the discharge of gas in the resin cavity.

#### 3.1.3. Effect of Process Conditions

During the process of pouring the resin into the end fitting cavity, the pressure differences between different parts of the resin drive the flow of the resin. Thus, increasing the pouring pressure facilitates the flow of the resin in narrow areas and improves the pouring quality. However, the pressure that pouring equipment can provide is limited, and the pressure that the end fitting resin cavity can withstand is also limited. Therefore, the smaller the pouring pressure is during the constant speed pouring process, the greater the additional pouring pressure that can be added, and the better the final pouring effect that can be obtained. The number of air traps that form directly indicates the pouring quality. Figure 7 shows the maximum pouring pressure at the gate and the number of air traps that form as a function of pouring rate and resin viscosity.

As shown in Figure 7a, the maximum pressure at the gate increases significantly with increasing pouring rate because increasing the pouring rate leads to an increase in shear rate, and shear stress increases as the shear rate increases, leading to an increase in resistance during the resin flow process; therefore, the pouring pressure required also in-creases. However, with increasing pouring rate, the number of air trap locations increases only slightly. This is because the viscosity of the resin is relatively high, making the influence of the viscous force greater than that of the inertial force during the pouring process. Therefore, within the range of the simulated pouring rates, changing the pouring rate does not significantly change the flow of the resin, so the number of air trap locations changes only slightly.

As shown in Figure 7b, the maximum pressure at the gate increases significantly with increasing resin viscosity. When the pouring rate remains unchanged, the shear rate of the resin does not change very much, while the shear stress increases with increasing resin viscosity, causing the resin flow resistance and the required pouring pressure to increase. With increasing resin viscosity, the number of air trap locations increases slightly. As mentioned above, most of the air traps generated during the pouring process are located in narrow areas in which it is difficult for the resin to flow; even when the resin viscosity is low, these areas are difficult to fill. Therefore, a slight change in resin viscosity does not result in a significant change in the number of air traps.

### 3.2. Effect of Tensile Armor Steel Wire on Pouring Quality

It can be seen from the simulation results and their analysis that the tensile armor steel wires greatly affect the resin pouring quality of the end fitting resin cavity. Therefore, some local pouring simulations were conducted in which the influence of the geometrical characteristics of the tensile armor steel wires on the pouring quality were studied.

#### 3.2.1. Bending Pattern of Steel Wires

First, the influence of the bending pattern of the steel wire on the pouring quality was studied.

Because the whole resin cavity contains a large number of steel wires, the situation is too complex to permit analysis. To facilitate the analysis, only a single steel wire and the part of the cavity that surrounds it are considered in the local simulation. This three-dimensional local resin flow was also simulated by using Moldflow. Four combinations of bending angles, 20°–6°, 25°–12°, 30°–18° and 35°–24°, were tested.

Figure 8 shows the resin filling process in the single steel wire pouring simulation when the combination of bend angles is 25°–12°. The total resin filling process takes 85.55 s, and ten time points during this period are shown. To observe the effect of the steel wire on the resin flow, the model was split along the axis of symmetry. As seen from the filling situations at 30.356 s and 39.555 s, due to the obstruction by the steel wire, a lag in resin filling occurred in the area below the bend of the steel wire; this may have led to failure to discharge some of the air present in this area in time, and air traps thus formed. At 67.152 s and 76.351 s, observation of the area in the lower right corner of the model clearly shows that the flow of resin in the narrow area lags behind that in the open area. This situation will result in filling the open area around the narrow area before the narrow area is filled with resin and will make it difficult for air to escape from the narrow area; thus, air traps will form. This is consistent with the result obtained in the pouring simulation.

Figure 9 shows the variation in the maximum pouring pressure at the gate and the maximum shear rate at different combinations of steel wire bending angles, i.e., as a function of changes in the bending pattern of the steel wire. As shown in Figure 9, as the bending angle of the steel wire is increased, the pouring pressure and the shear rate show an increasing trend. Compared with the pouring pressure observed in the first three groups of bending angles, the pouring pressure and shear rate in the last group of bending angles increase significantly. This is because the distance between the middle part of the steel wire and the top wall is too small, resulting in a sudden increase in the resistance of the axial flow of the resin above the steel wire and a sudden increase in the pouring pressure. Moreover, the increased resistance in the resin axial flow leads to an increase in the resin flow from the area above the steel wire to the area below the steel wire, consistent with the sudden increase in the maximum shear rate. Figure 10 shows a comparison of the filling situation when the resin flows over the top of the steel wire when the bending angle combination of the steel wire is 35°–24° with that when the bending angle combination is 30°–18°. At the moment when the resin just begins to flow over the top of the steel wire, the cavity filling rate is greater when the bending angle combination is 35°–24° than it is when the bending angle combination is 30°–18°, and the proportion of resin flowing below the steel wire is greater. This indicates that the axial flow of the resin above the steel wire is more difficult; under these circumstances, more resin is forced to flow below the steel wire, consistent with the previous analysis.

#### 3.2.2. Steel Wire Section Shape

In addition, some two-dimensional simulations of the resin flow around the steel wire were conducted to study the influence of the steel wire section shape on the pouring quality. The Laminar Two-Phase Flow Phase Field Interface of Comsol was used for simulation.

Round steel wire sections with a radius of 2.5 mm and three types of rounded rectangular steel wire sections with different corner radii were modeled; the three different corner radii of the rounded rectangle were 0 mm, 0.3 mm and 0.6 mm.

Figure 11 illustrates the resin flow around the steel wire for a corner radius of 0.3 mm. The total duration of the simulation is 15 s, and nine time points during this period are shown in the figure. It can be seen from Figure 11 that after the resin flowed over the surface of the steel wire, a small amount of air remained on the rear flow surface of the steel wire, forming air traps. In the pouring simulation of the resin cavity, such air traps are also observed, but the resin flow along the steel wire direction in the three-dimensional simulation is beneficial in eliminating such air traps. Therefore, the number of such air traps observed in the pouring simulation of the resin cavity is small. In addition, it can be seen that during the process of resin flow, small bubbles will occur.

Figure 12 compares the air traps retained at the rear flow surface at 15 s when different cross-sectional shapes of the steel wire are used. When the wire cross-section is rectangular or rounded rectangular, the amount of air retained decreases slightly as the radius of the rounded corner increases, whereas when the wire cross-section is rounded, the amount of air retained decreases considerably. It can be seen that increasing the radius of the rounded corners of the wire cross-section helps improve the pouring quality when the steel wire cross-section is rounded rectangular, while the use of a wire with a rounded cross-section shape is more beneficial than the use of a flat wire with a rectangular or rounded rectangular cross-section.

### 3.3. Validation of Results

According to the obtained simulation results, it is concluded that the use of a cavity wall with a smooth transition, a lower pouring rate, a lower resin viscosity, a steel wire bending pattern with lower middle uplift and a round steel wire are conducive to improvement of the pouring quality. Based on this, the resin cavity structure and pouring process were modified, and a pouring simulation was performed to verify the conclusion. The pouring rate was 40 cm^3^/s, the reference viscosity of the resin was 30 Pa·s, the bending angle of the steel wire was 20°–6°, and the diameter of the round steel wire was 4.14 mm. The proper position of each component of the end fitting was chamfered. A wire diameter of 4.14 mm was chosen to ensure that the contact area between the wire and the resin in the cavity remained approximately constant.

In the final simulation, the maximum pouring pressure at the gate was reduced to 0.9 KPa. The resulting air traps are shown in Figure 13. On the one hand, the distribution of the air traps does not change significantly; the vast majority of the air traps are still located on the surface of the steel wire, and most of these air traps are still located where the two layers of armor steel wires overlap. On the other hand, although the number of air traps did not decrease significantly, it can be seen that the volume of the individual air traps decreased significantly.

## 4. Conclusions

In this paper, to solve the problem of a low filling rate of the end fitting resin cavity of a deep-water composite flexible pipe, the resin pouring process of the end fitting resin cavity was studied by numerical simulation. Through the pouring simulation of the end fitting resin cavity, the distribution of air traps generated during the resin pouring process was confirmed. Most of the air traps are located on the surfaces of the steel wires, and a small fraction of them are located at the corner of the cavity. Furthermore, the air traps on the surfaces of the steel wires are mostly located in the narrow areas created by the presence of the steel wires, such as in the areas in which two layers of armor steel wires overlap and the areas where the steel wires are gathered. In addition, it is observed in the simulations that because of the high viscosity of the resin and the obstruction of resin flow by the steel wire, part of the resin flows above the steel wires, and the other part flows below the steel wires. This may lead to the formation of air traps, as the upper layer of resin may block the discharge of air from below when the resin flows in the axial direction. In the narrow area of the cavity, difficulty in resin filling is observed. The effects of pouring rate and resin viscosity on pouring quality were also studied. The results show that use of a low pouring rate and a low resin viscosity are beneficial to the improvement of pouring quality. Study of the influence of the bending pattern and the cross-section shape of the armor steel wires on the pouring quality shows that the use of a steel wire bending pat-tern with lower middle uplift is more conducive to obtaining good pouring quality, while in terms of cross-sectional shape, a rounded rectangle with a larger corner radius is better than a rounded rectangle with a smaller corner radius, and round steel wire is better than flat steel wire. Finally, based on the results obtained, the end fitting structure and the process conditions were optimized; under the optimized conditions, the maximum pouring pressure at the gate was reduced to 0.9 KPa, and the volume of air traps was significantly reduced.

## Figures and Tables

**Figure 1 polymers-15-01397-f001:**
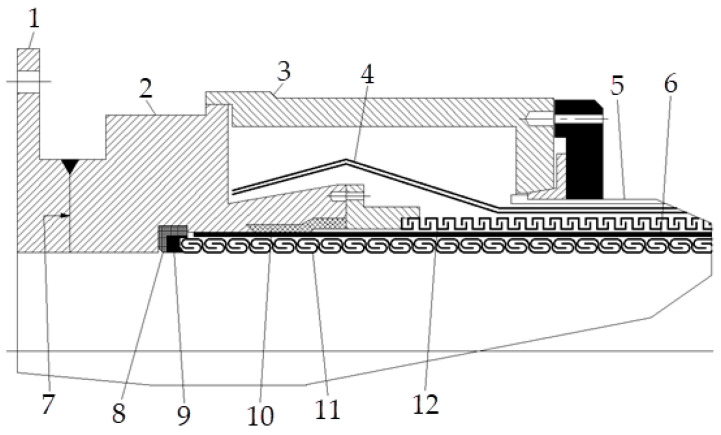
Typical structure of a flexible pipe end fitting. 1: mounting flange; 2: inner casing; 3: outer casing; 4: tensile armor layer; 5: outer sheath; 6: inner sheath; 7: end fitting neck; 8: insulator; 9: carcass end ring; 10: sealing ring; 11: carcass; 12: pressure armor layer.

**Figure 2 polymers-15-01397-f002:**
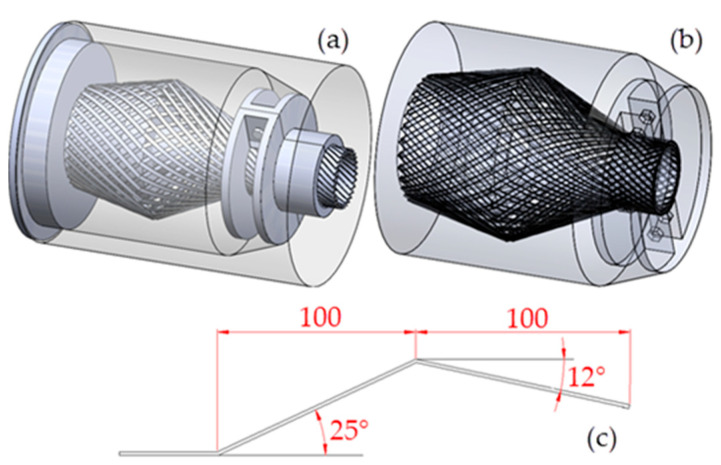
Geometric models used in pouring simulation: (**a**) end fitting model; (**b**) resin cavity model; (**c**) steel wire bending pattern.

**Figure 3 polymers-15-01397-f003:**
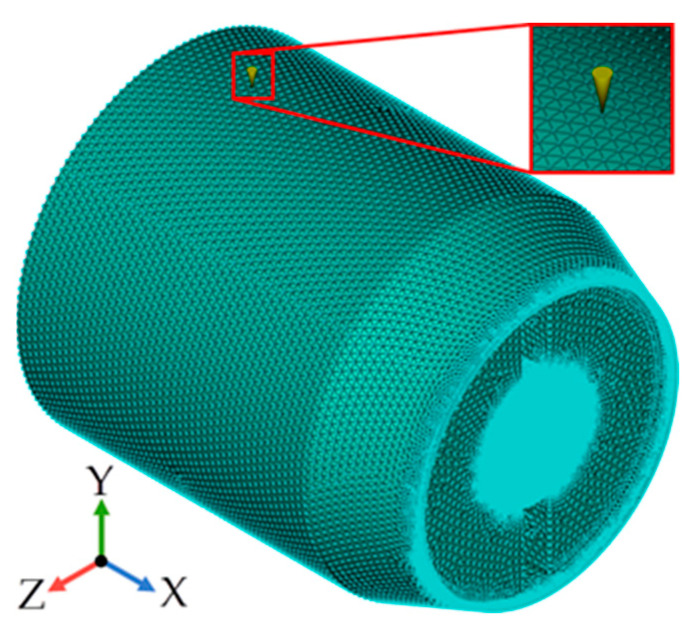
Finite element model of the resin cavity.

**Figure 4 polymers-15-01397-f004:**
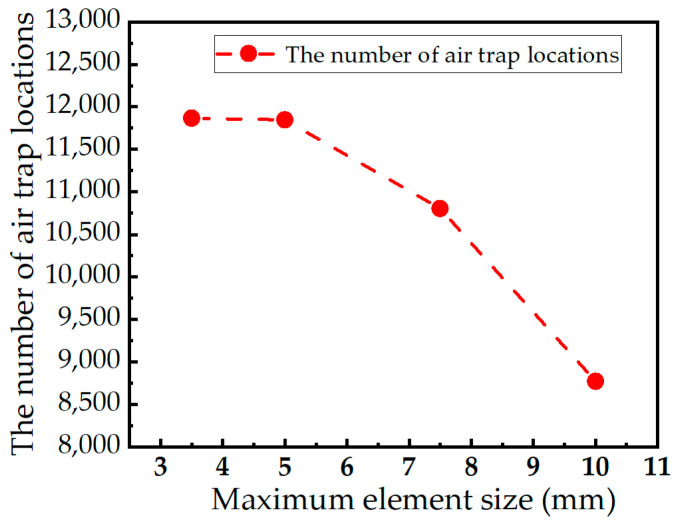
Variation in the number of air trap locations as a function of maximum element size.

**Figure 5 polymers-15-01397-f005:**
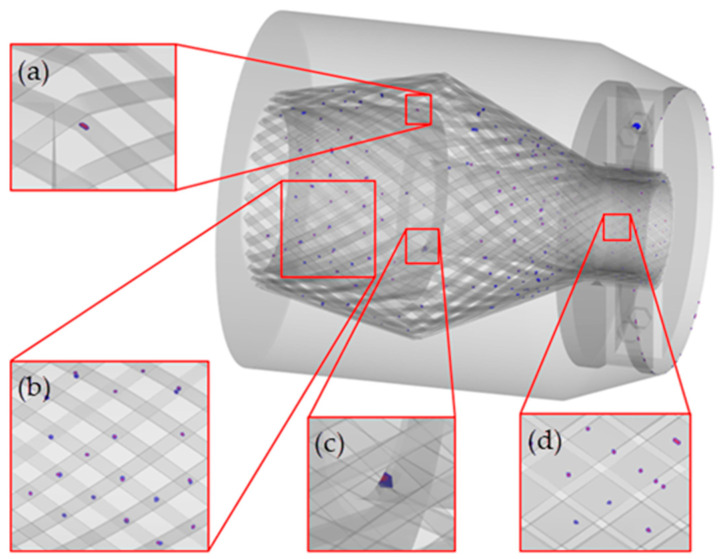
Air trap distribution results: (**a**) bending of the steel wires; (**b**) overlap of the steel wires; (**c**) corner of the cavity wall; and (**d**) gathering of the steel wires.

**Figure 6 polymers-15-01397-f006:**
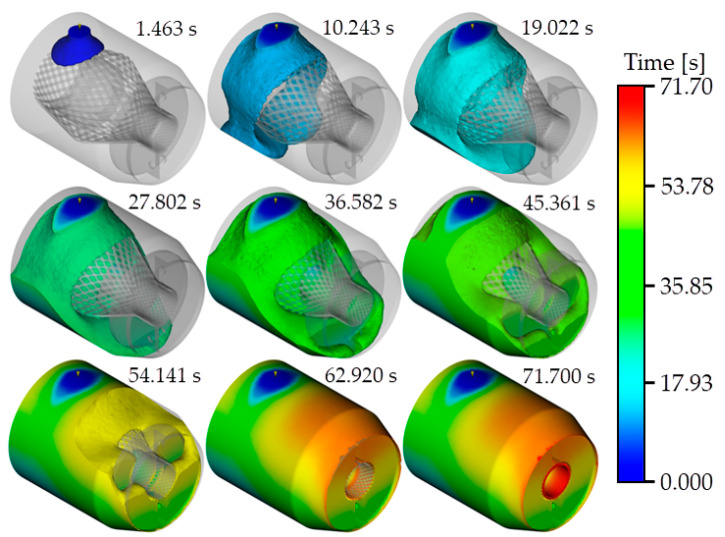
Resin filling process in the pouring simulation.

**Figure 7 polymers-15-01397-f007:**
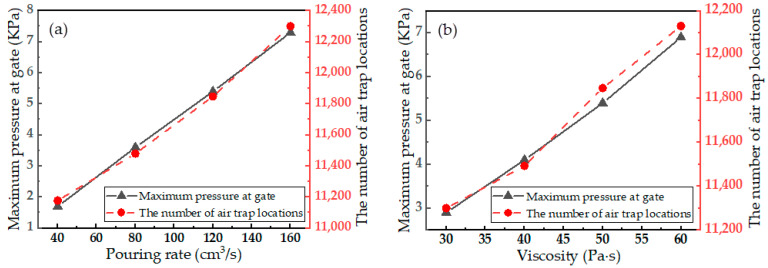
Variation in maximum pouring pressure and number of air trap locations as a function of (**a**) pouring rate and (**b**) resin viscosity.

**Figure 8 polymers-15-01397-f008:**
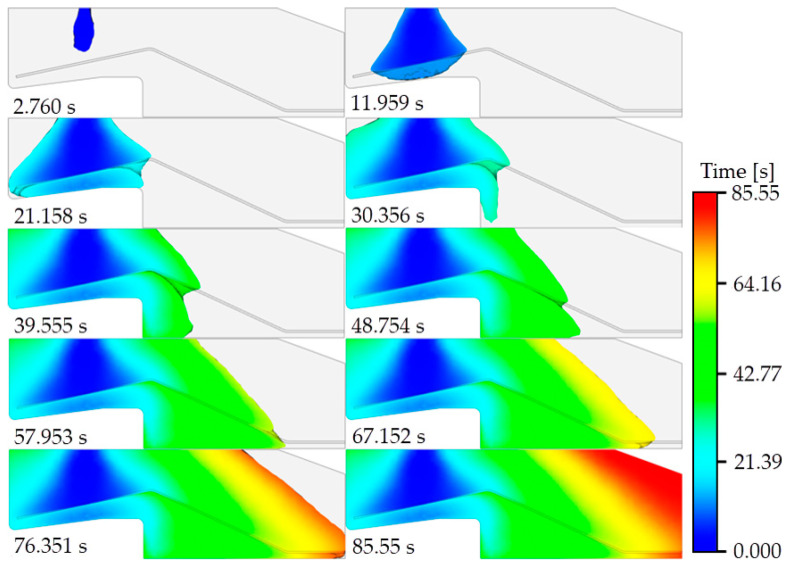
The resin filling process in the single steel wire pouring simulation.

**Figure 9 polymers-15-01397-f009:**
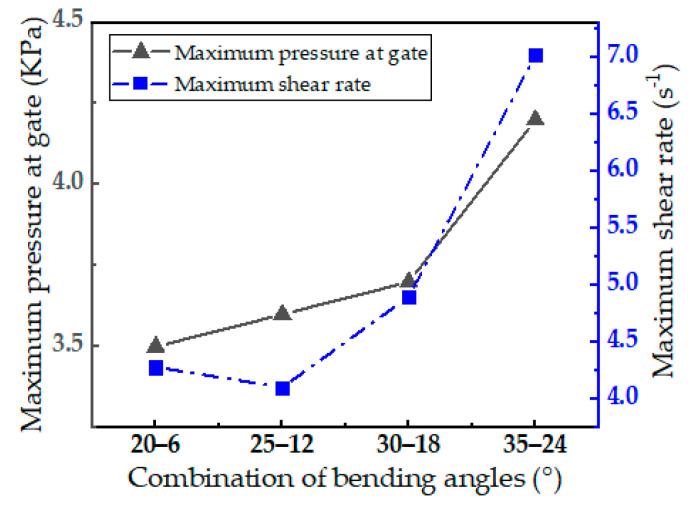
Variation in the maximum shear rate and the maximum pouring pressure at the gate with changes in the combination of the bending angle.

**Figure 10 polymers-15-01397-f010:**
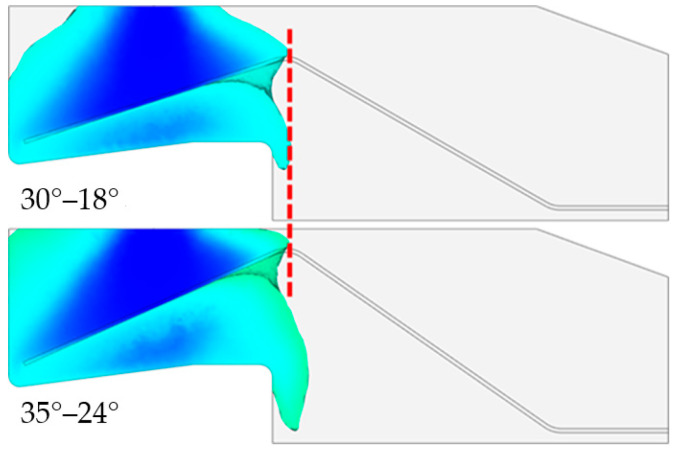
Comparison of the filling situation when the resin flows over the top of the steel wire at steel wire bending angle combinations of 35°–24° and 30°–18°.

**Figure 11 polymers-15-01397-f011:**
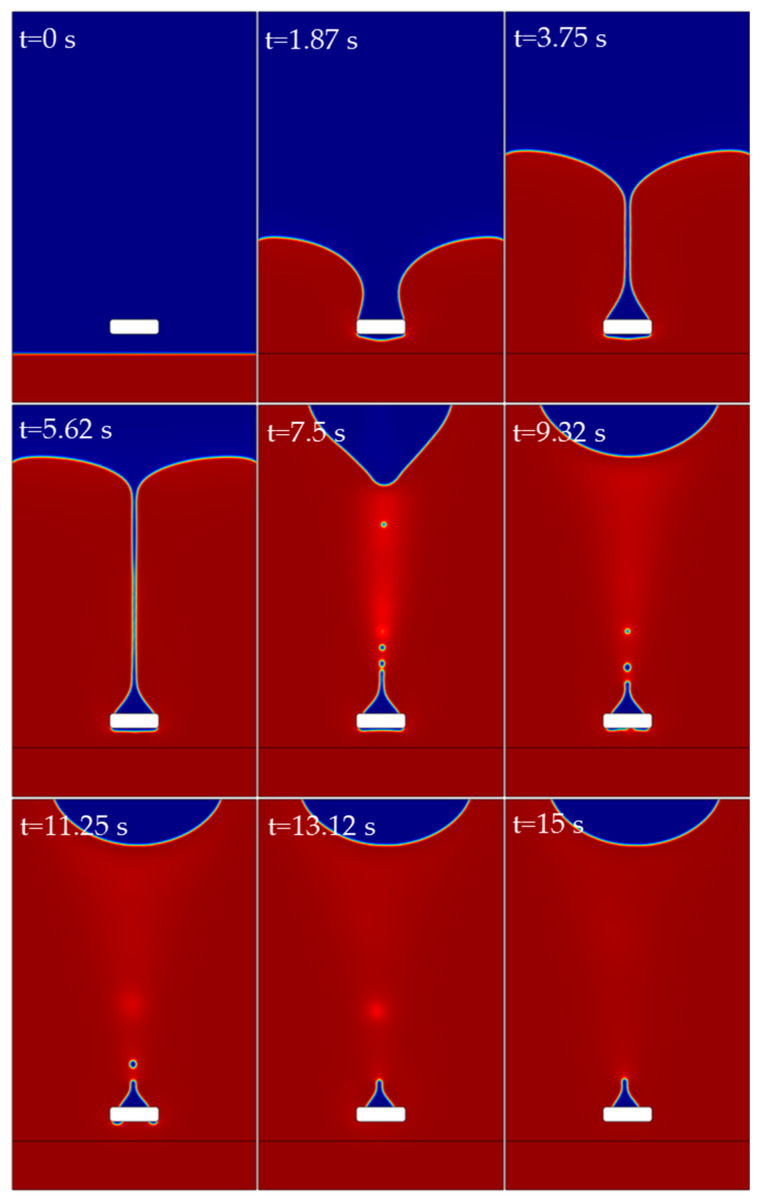
Resin flow around the steel wire when the corner radius is 0.3 mm.

**Figure 12 polymers-15-01397-f012:**
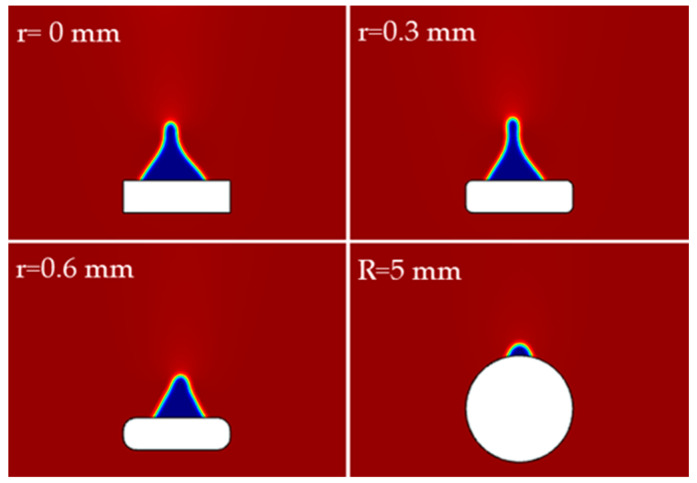
Gas residue when steel wires with different cross-sectional shapes are used.

**Figure 13 polymers-15-01397-f013:**
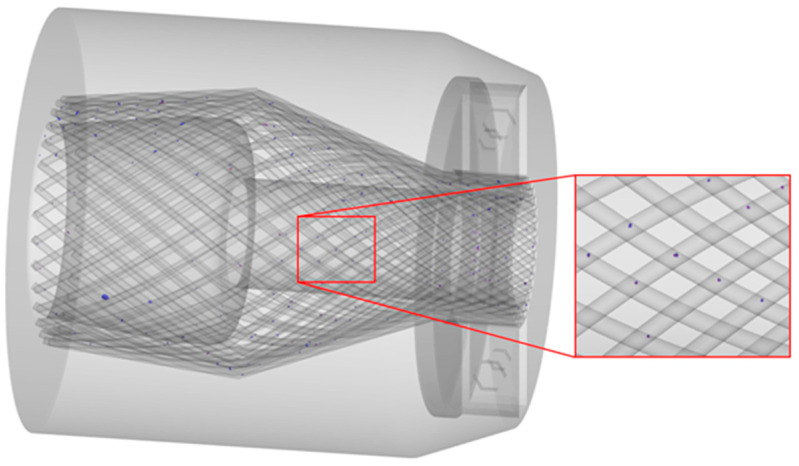
Air trap results after optimization of the structure and process conditions.

## Data Availability

Not applicable.

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
