# Peer review of "Numerical Simulation of the Pouring Process in Deep-Water Flexible Pipe End Fittings"

_polymers, 2023, doi:10.3390/polym15061397_

Round 1
Reviewer 1 Report
Only three minor changes should be made.
Line 83: Use the abbreviation introduced in Line 79.
Line 116: Section 2 should be renamed as "Methods".
Lines 307-308: The end of this sentence needs rephrasing.
Reviewer 2 Report
1. The article does not contain information on what software was used for numerical simulations.
2. Is the epoxy resin model data from a specific resin manufacturer or is it literature data?
Reviewer 3 Report
according my review of the reference paper, we found a research of Numerical simulation of the pouring process. I read the concept, boundary conditions and results. But the authors don't answer the following questions:
What is the simulation validation method?
What's the quality of simulation results?
What's the accuracy/errors found in the simulation results vs experimental filling/curing parts?
Have the errors been statistically studied? (like voids and air traps compared between simulaton and reality)
No comparative real results were found to establish the simulation errors. What is the difference with the real resin behaviour?
Round 2
Reviewer 3 Report
paper can be accepted as is